# Development of Liposome Systems for Enhancing the PK Properties of Bivalent PROTACs

**DOI:** 10.3390/pharmaceutics15082098

**Published:** 2023-08-08

**Authors:** Ponien Kou, Elizabeth S. Levy, An D. Nguyen, Donglu Zhang, Shu Chen, Yusi Cui, Xing Zhang, Fabio Broccatelli, Jennifer Pizzano, Jennifer Cantley, Elizabeth Bortolon, Emma Rousseau, Michael Berlin, Peter Dragovich, Vijay Sethuraman

**Affiliations:** 1Small Molecules Pharmaceutics, Genentech, 1 DNA Way, South San Francisco, CA 94080, USA; koup@gene.com (P.K.); levye3@gene.com (E.S.L.); anguyen@ideayabio.com (A.D.N.); 2Drug Metabolism & Pharmacokinetics, Genentech, 1 DNA Way, South San Francisco, CA 94080, USA; zhangd38@gene.com (D.Z.); chens171@gene.com (S.C.); cuiy23@gene.com (Y.C.); zhanx239@gene.com (X.Z.); fabio.broccatelli@bms.com (F.B.); 3Arvinas LLC, 5 Science Park, New Haven, CT 06511, USA; jennifer.pizzano@arvinas.com (J.P.); jennifer.cantley@arvinas.com (J.C.); elizabeth.bortolon@arvinas.com (E.B.); emma.rousseau@arvinas.com (E.R.); michael.berlin@arvinas.com (M.B.); 4Medicinal Chemistry, Genentech, 1 DNA Way, South San Francisco, CA 94080, USA; dragovip@gene.com

**Keywords:** PROTAC, liposome, nanoparticle, drug delivery

## Abstract

Proteolysis-Targeting Chimeras (PROTACs) are a promising new technology in drug development. They have rapidly evolved in recent years, with several of them in clinical trials. While most of these advances have been associated with monovalent protein degraders, bivalent PROTACs have also entered clinical trials, although progression to market has been limited. One of the reasons is the complex physicochemical properties of the heterobifunctional PROTACs. A promising strategy to improve pharmacokinetics of highly lipophilic compounds, such as PROTACs, is encapsulation in liposome systems. Here we describe liposome systems for intravenous administration to enhance the PK properties of two bivalent PROTAC molecules, by reducing clearance and increasing systemic coverage. We developed and characterized a PROTAC-in-cyclodextrin liposome system where the drug was retained in the liposome core. In PK studies at 1 mg/kg for GNE-01 the PROTAC-in-cyclodextrin liposome, compared to the solution formulation, showed a 80- and a 380-fold enhancement in AUC for mouse and rat studies, respectively. We further investigated the same PROTAC-in-cyclodextrin liposome system with the second PROTAC (GNE-02), where we monitored both lipid and drug concentrations in vivo. Similarly, in a mouse PK study of GEN-02, the PROTAC-in-cyclodextrin liposome system exhibited enhancement in plasma concentration of a 23× increase over the conventional solution formulation. Importantly, the lipid CL correlated with the drug CL. Additionally, we investigated a conventional liposome approach for GNE-02, where the PROTAC resides in the lipid bilayer. Here, a 5× increase in AUC was observed, compared to the conventional solution formulation, and the drug CL was faster than the lipid CL. These results indicate that the different liposome systems can be tailored to translate across multiple PROTAC systems to modulate and improve plasma concentrations. Optimization of the liposomes could further improve tumor concentration and improve the overall therapeutic index (TI). This delivery technology may be well suited to bring novel protein targeted PROTACs into clinics.

## 1. Introduction

Cancer is one of the leading causes of death, with rapidly growing worldwide incidences and mortality [1]. Although numerous therapies have been developed to treat cancer, the worldwide burden has consistently grown. An important gene, BRM (Brahma homolog; also known as SWI/SNF-related, matrix-associated, actin-dependent regulator of chromatin subfamily A member 2 or SMARCA2), has been identified as a promising new target in promoting tumor suppression. The BRM protein is an essential catalytic component of the SWI/SNF chromatin remodeling complex, which is known to play a key role in cancer progression [2]. Techniques to inhibit, downregulate, and degrade the BRM protein have, therefore, been explored to enable the development of new cancer therapies [2,3].

An emerging strategy for targeted treatment of various diseases, including cancer, has been Proteolysis-Targeting Chimeras (PROTACs). PROTACs have been successful in early clinical trials to efficiently degrade a targeted protein in prostate and breast cancers [4]. They have many advantages over traditional small molecule inhibitors, including a possible catalytic mechanism of action resulting in low required concentrations to elicit a therapeutic response [4,5]. Another advantage is that PROTAC molecules only require binding to their targets as opposed to traditional small molecules, which typically need to inhibit (or modulate) biological activity. Hence, chemical design can be more focused on binding affinity rather than inhibition. This attribute also opens therapeutic possibilities for the >80% proteins that are currently considered undruggable with conventional small molecules and antibodies [6].

There are two kinds of protein degraders, namely monovalent [7] and heterobifunctional [4] degraders. Monovalent protein degraders can be further divided into two categories: (A) a molecule that binds to an E3 ligase and, thereby, enables formation of a ternary complex with the target protein (resulting in its subsequent degradation; often termed a “molecular glue”) and (B) a molecule that can bind directly to the target protein with subsequent degradation occurring via several possible mechanisms (e.g., conformational changes, lysine mimicry, or E3 ligase complex formation; in the latter case, the degraders do not directly associate with the ligase in the absence of the target protein). However, optimization of both the monovalent degrader classes is highly challenging, with a structure–activity relationship driven by mostly empirical methods. The other type of protein degrader, the heterobifunctional molecule, consists of two ligands connected by a linker molecule. One ligand binds to the target protein, while the other ligand interacts with an E3 ligase that functions to recruit the ubiquitin machinery to the target protein leading to its degradation [4]. Although the heterobifunctional PROTACs are larger in size, compared to the monovalent degraders, the heterobifunctional PROTACs have potential for more specific interaction with active recruitment of E3 ligase for targeted degradation.

The first heterobifunctional PROTAC was introduced in 2001, targeting methionine aminopeptidase-2 [8]. The field has since rapidly developed, with the discovery of new E3 ligase ligands and expansion of the chemical space to allow for not only parenteral, but also oral, delivery, with two orally delivered PROTACs in clinical trials [9]. Although there has been significant development in the chemical matter space, many of the heterobifunctional PROTACs have high molecular size and high lipophilicity, resulting in poor permeability and solubility compared to conventional small molecules [10,11,12]. These exigent physicochemical properties have led to challenges in formulation development and biological activity.

Over the years there has been an increase in the proportion of new molecules that are poorly water-soluble [13]. These highly lipophilic and low aqueous solubility molecules can be challenging to formulate, especially as intravenous (I.V.) drug products. Generally, in the I.V. dosage form, the drug is administered in a soluble state. This may limit the dose ranges of poorly soluble compounds and pose an increased risk of precipitation in vivo [14].

One strategy to enhance solubility of highly lipophilic molecules is the utilization of cyclodextrins. Cyclodextrins have a unique cyclic structure that includes a hydrophobic core and a hydrophilic exterior. This design allows them to house hydrophobic molecules within the core; thus, enhancing the water-solubility of the molecules that are typically water-insoluble [15,16]. Numerous derivatives of cyclodextrins have been researched, each offering unique size, shape, and water solubility characteristics that result in various complexation interactions with drug molecules [17]. While cyclodextrins improve solubility, they generally do not alter the pharmacokinetic properties of the compound [18]. Aside from investigating cyclodextrins as a means to improve water solubility, it is crucial to develop alternative formulation techniques to improve the PK properties of insoluble compounds as well.

Liposome drug carriers have been shown to improve both pharmacokinetic properties and efficacy of small molecule therapeutics [19]. Liposomes are very versatile and can encapsulate both lipophilic and hydrophilic therapeutics in the lipid bilayer and aqueous core, respectively. This encapsulation protects the drug during systemic circulation; thus, increasing the systemic half-life, and improving the therapeutic index [20]. In addition to traditional liposomes, cyclodextrin-in-liposome systems have also been developed where the hydrophobic molecule localizes within the aqueous core of the liposome, rather than in the lipid bilayer. Localizing the compound in different regions of the liposomes can result in different release profiles [21]. Nanoparticle carriers, based on polymers and liposomes, have been investigated as promising strategies to deliver PROTACs, due to their lipophilic natures and the intracellular localization required to generate a therapeutic response [15,22]. PROTACs are generally dosed by means of I.V., due to their poor oral bioavailability, so development of strategies to increase the systemic half-life can reduce the dosing frequency.

Here, we discuss the investigation of a conventional, and a cyclodextrin-based, liposome system encapsulating the BRM target heterobifunctional PROTACs, GNE-01 and GNE-02; mainly to improve systemic clearance. Both these molecules are highly lipophilic and are rapidly cleared from systemic circulation; hence, an alternative strategy to conventional solution formulation was required to reduce clearance and increase drug deposition. We developed and characterized liposomal formulations for the PROTACs and determined their pharmacokinetics (PK) in both mouse and rat models. We also characterized the pharmacodynamic effect of the liposomal formulation in a BRM-relevant mouse tumor model. To our knowledge, this is the first study to compare different liposome formulations to alter the PK of PROTACs. Development of these liposome technologies can enable delivery of PROTAC molecules with optimal PK properties, enhancing their therapeutic responses.

## 2. Experimental

### 2.1. Materials

The lipids, including 1,2-distearoyl-sn-glycero-3-phosphocholine (DSPC), 1,2-distearoyl-sn-glycero-3-phosphoethanolamine-N-[methoxy(polyethylene glycol)-2000] (DSPE-PEG2000), and cholesterol, were purchased from Avanti (Birmingham, AL, USA). Ethanol was purchased from JT. Baker (Philadelphia, PA, USA). The (2-Hydroxypropyl)-β-cyclodextrin (HP-β-CD) and Sodium Acetate were purchased from Sigma-Aldrich (St. Louis, MO, USA). Acetic acid and dextrose were purchased from EMD Millipore (Billerica, MA, USA). The dialysis cassettes were purchased from Fisher Scientific (Newington, NH, USA). The AKTA column was a HiPrep 26/10 Desalting column and purchased from Cytiva (Marlborough, MA, USA). The TFF column was MidiKros 20 cm 300 k MPES 0.5 mm (Part Number: D02-E300-05-N) and purchased from Repligen (Waltham, MA, USA).

### 2.2. Conventional Liposome Preparation

The conventional liposomes were prepared through a microfluidics method with the NxGen Blaze (Precision NanoSystems, Vancouver, BC, Canada) system, as previously described in [23]. First, the DSPC, DSPE-PEG2000, and cholesterol (61.56%:18.8%:19.56% (*w/w*)) were dissolved, at a total lipid concentration of 30 mg/mL, in ethanol. GNE-02 was dissolved in the above lipid solution at 0.5 mg/mL. This ethanol solution was rapidly mixed with 50 mM sodium acetate buffer solution at pH 4.0 at a 3:1 (aqueous:organic) Flow Rate Ratio (FRR) through the NxGen Blaze with classic blaze cartridge (part number: NIB0002) at a Total Flow Rate (TFR) of 12 mL/min. The liposome product formed from the NxGen Blaze was purified by tangential flow filtration (TFF) (KrosFlo KR2i TFF System) to remove ethanol and any unencapsulated drug, using the sodium acetate buffer for buffer exchange. In the TFF cycle, the liposomes were concentrated 2.4× and dia-filtrated 6× to the final concentration of 0.9 mg/mL of GNE-02. Then, the liposomes were injected into an AKTA-Pure chromatography system (GE Healthcare, Amersham, England) with an HiPrep 26/10 Desalting column for further purification at 5 mL/min flow rate, and pure liposome fractions were collected by a Frac 920 collector.

### 2.3. PROTAC-in-cyclodextrin Liposome Preparation

The PROTAC-in-cyclodextrin were prepared with a similar technique as conventional liposomes with the following difference. The DSPC, DSPE-PEG2000, and cholesterol (61.56%:18.8%:19.56% (*w/w*)) were dissolved at a total lipid concentration of 200 mg/mL in ethanol at 60°C. The drug was dissolved in 20% HP-β-CD, 5% Dextrose aqueous solution at 25 mg/mL concentration, instead of being dissolved in ethanol, as with the above conventional liposome preparation. The organic lipidic solution was mixed with the aqueous solution at the same FRR and TFR as conventional liposomes through a Nanoassemblr (Precision NanoSystems, Vancouver, BC, Canada) system, with a heating controller accessory at 60 °C. The resulting PROTAC-in-cyclodextrin liposomes were centrifuged at 2000× *g* for 10 min to remove any precipitated debris, and the supernatant was purified by dialysis with 5% dextrose and buffer exchanged at 2, 4 and 12 h. Then, the sample was collected after 24 h.

### 2.4. Solution Formulation Preparation

First, a 50 mM sodium acetate buffer of pH 4.0 was prepared and, to this, 20% *w/v* of HP-β-CD was slowly added, while stirring, to obtain a clear solution. This vehicle was used to fully dissolve the PROTAC to the target concentration at ambient temperature by stirring overnight.

### 2.5. Characterization of Liposomes

The particle size distribution and polydispersity index (PDI) were characterized by dynamic light scattering (DLS), using a Malvern Zetasizer Ultra (Malvern Instruments, Malvern, UK). Liposomes were diluted 5 to 10 times, depending on the starting concentration, into appropriate buffers and then 1 mL of sample was transferred to a cuvette for measurement. We also analyzed zeta potential, based on the principles of laser Doppler velocimetry and electrophoretic mobility (EM) with the same instrument. All readings were taken at 25 °C. Data analysis was performed by means of ZS Xplorer software (version 1.0).

### 2.6. Encapsulation Efficiency (EE) Determination

We refer to encapsulation efficiency as an expression of drug in the liposome formulation relative to the total amount of drug in the starting materials. The higher the encapsulation efficiency, the more drug remains in the liposome formulation during the manufacturing and purification processes. Both the total amount of drug and dosing concentration were measured by HPLC. The encapsulation efficiency was expressed as the percent of drug encapsulated and calculated by the following equation: (1)$$\mathrm{Encapsulation}\;\mathrm{Efficiency}=\frac{\mathrm{API}\;\mathrm{Concentration}\;\mathrm{in}\;\mathrm{Liposome}}{\mathrm{Total}\;\mathrm{Amount}\;\mathrm{of}\;\mathrm{API}\;\mathrm{in}\;\mathrm{Starting}\;\mathrm{Material}}\times 100\%$$

We define the loading efficiency as the percent of encapsulated drug over the total drug in the dosing formulation, which was calculated with the following equation: (2)$$\mathrm{Loading}\;\mathrm{Efficiency}=\frac{\mathrm{Encapsulated}\;\mathrm{API}}{\mathrm{Total}\;\mathrm{Amount}\;\mathrm{of}\;\mathrm{API}\;\mathrm{in}\;\mathrm{Formulation}}\times 100\%$$

### 2.7. Log P/pKa

Log P was measured by the shake flask method using a high throughput Tecan robot. Briefly, octanol was added into a 96 well plate and the drug was spiked into the octanol solution. Into this, water (Log P) or acetate buffer at pH 4.0 (Log D) was added. The mixture was then shaken for 5 min. The plate was then centrifuged for 10 min at 3700 rpm to separate the octanol and water layer. Samples from each layer were taken and analyzed by means of LCMS (Sciex 7500 (Framingham, MA, USA)).

### 2.8. Pharmacokinetics (PK) of PROTAC and Lipid Analysis

Animal studies for PK and PK/PD were approved by the Institutional Animal Care and Use Committee (IACUC) of both Genentech and Arvinas, in accordance with federal guidelines. All animal studies complied with the ethical regulations and humane endpoint criteria, according to the NIH Guidelines for the Care and Use of Laboratory Animals.

For GNE-01 PK study, a total of 6 Sprague Dawley rats, weighing 250 to 310 g, were acquired from Charles River Labs (Wilmington, MA, USA) and divided evenly into 2 treatment groups (all males). Each group was intravenously administered 1 mg/kg of either PROTAC-in-cyclodextrin liposome or conventional solution formulation (20% HP-β-CD in acetate buffer at pH 4.0) by I.V. bolus through a jugular vein cannula. Blood collection was performed at the following time points: pre-dose, 0.033, 0.083, 0.25, 0.5, and 1, 2, 4, 8, 24, 48 and 96-h post-dose. All samples were collected through a femoral artery cannula by a Culex automated blood sampling machine (West Lafayette, IN, USA) and placed in tubes containing K2 EDTA. Blood samples were centrifuged for 5 min at 3700 rpm (2235 g) for plasma collection. Similar PK studies were performed in SCID mice as well at n = 3 per arm.

Drug plasma concentrations were quantified via a liquid chromatographic-tandem mass spectrometry (LC/MS/MS) assay method with an internal standard. PK parameters for blood and organ tissues were determined by non-compartmental methods using the IV-bolus input model, Phoenix™ WinNonlin, version 6.4 (Certara USA, Inc., Princeton, NJ, USA).

For GNE-02 PK, a total of 33 CD-1 male mice weighing 18 to 25 g were acquired from Charles River Labs (Wilmington, MA, USA) and divided into 4 treatment groups. The firsst group was the conventional liposome group with 15 animals, the second group was the PROTAC-in-cyclodextrin liposome group with 3 animals, and the third group was the conventional solution formulation arm. The liposome and solution arms, with 15 animals, were required for organ quantification. Sparse sampling was performed for tissue and blood samples. The mean time concentration profile was generated for each group and an overall AUC was then calculated with these sparse samples. The dosing and post-dose analysis was performed similar to GNE-01.

For lipid analysis, an HILIC column (Kinetex 2.5 µm HILIC 2.1 × 50 mm, Phenomenex, Torrance, CA, USA) was used. Mobile phase A was water (with 0.1% formic acid) and B was acetonitrile (with 0.1% formic acid). Mass spectrometry data was acquired using a Thermo Fisher Q-Exactive HFX in targeted SIMS mode with a resolving power of 60,000, AGC target of 2 × 10${}^{5}$, maximum injection time of 100 ms and isolation window of 4 *m*/*z*. The SIMS inclusion list contained the lipid ion (*m*/*z* = 790.63203, z = 1) and loperamide ion (*m*/*z* = 477.23033, z = 1).

All the RAW data was processed using Excalibur software (version 2.2SP1). The peak was detected by means of the ICIS algorithm. Sciex WIFF Data was processed by means of Analyst software (Elucidator software (version 3.3.0.1 SP4.25)) and peaks were detected by means of the MQ4 algorithm. Quadratic calibration curves of analyte to IS peak area ratio was fitted over concentration with the weighing power of 1/x, and the lipid and payload concentration in the unknown samples was calculated based on the calibration curve. The LLOQ of payload was 45.7 ng/mL. The LLOQ of lipid was higher at 412 ng/mL, due to the interference from a high level of endogenous lipid.

### 2.9. Pharmacodynamics (PD)

Calu6 tumors (BRG1 wild-type) were selected for the PK/PD experiments. Calu6 cells were purchased from ATCC (HTB-56) and cultured in EMEM with 1% Penicillin/streptomycin, 1% HEPES, and 10% FBS in a humidified incubator at 37 °C and 5% CO_2_.

Female Crl:NU-Foxn1nu (NU/NU Nude) mice, aged 6–8 weeks, were used for the xenograft model and purchased from Charles River laboratories. The mice received food and water ad libitum and were allowed to acclimate for 1–2 weeks before being used for experiments.

Calu6 tumor cells were implanted into the right flank of the NU/NU SCID mice. The tumor growth was monitored daily, and tumors were measured twice a week using digital calipers. Tumor volume was determined using the following formula (width × width × length)/2), where all measurements were in mm and the tumor volume was in mm^3^. The treatment was started once the average tumor volume reached 150–200 mm^3^. Treatment was started approximately 3 weeks after cell implantation. The animals were randomly assigned into 3 groups (n = 6 animals per group), such that each group had nearly equal starting average tumor volume. Treatment groups were randomly assigned into groups treated with GNE-01 solution or PROTAC-in-cyclodextrin liposome. GNE-01 was dosed at 10 mg/kg of body weight into the lateral tail vein intravenously once for the PK/PD studies. All dosing solutions were filtered prior to injection using a 0.2-micron filter to ensure sterility. Post euthanasia, blood and various tissues, including tumors, were collected for further analyses.

The mice were euthanized at 96 h post-dosing. The euthanasias were performed following the IACUC approved method of euthanasia. Tumor tissues were collected and processed with western blot analysis for BRM inhibition. The western blot was performed to quantify the loss of BRM and BRG1 protein in the tumor tissue. An in-depth description of the PD methodology can be found elsewhere [24].

### 2.10. Statistical Analysis

Statistical analysis was performed using GraphPad Prism 8.0 Software (San Diego, CA, USA). A *t*-test with Welch correction or one-way analysis of variance (ANOVA) was used to analyze significance between each group. Data is expressed as Mean ± standard deviation (SD) where * indicates *p* < 0.05 and ** indicates *p* < 0.01. A non-compartmental analysis was conducted, utilizing the trapezoidal rule and relevant pharmacokinetics equations to calculate the pharmacokinetic parameters.

## 3. Results and Discussion

Two PROTAC compounds, GNE-01 and GNE-02, with similar physicochemical properties, as shown in Table 1, were investigated. Both compounds are highly lipophilic molecules with very limited water solubility (<1 µM). Development of I.V. formulations for compounds with low water solubility can be challenging and cyclodextrins have been used as solubilizers in pharmaceutical technology to enhance solubility of both oral and I.V. formulations. Cyclodextrins have been developed with different derivatives that consist of varying physicochemical properties influencing their aqueous solubility and capacity. One of the common natural cyclodextrins is β-cyclodextrin. However, β-cyclodextrin is not suitable for parenteral formulations, due to its limited solubility in water and associated toxicity [25]. In contrast, β-cyclodextrin derivatives, such as Sulfobutylether-β-Cyclodextrin (SBE-β-CD) and 2-Hydroxylpropyl-β-Cyclodextrin (HP-β-CD), possess higher aqueous solubility and have been successfully marketed in parenteral formulations, indicating a viable way forward with these derivatives. Through formulation screening, HP-β-CD was identified as the best cyclodextrin solubilizer for GNE-01 and the reagent was used for GNE-02 as well. In an aqueous solution of 20% HP-β-CD in D5W, both GNE-01 and GNE-02 solubility improved from <1 µM to >28 mM (Table 1).

In addition to the cyclodextrin solution formulation, two liposome formulation systems were also developed. Liposomes have demonstrated, both in research and clinical settings, to be a promising system for delivery of complex payloads via the parenteral route [26]. They are versatile carriers, wherein a lipophilic drug can be encapsulated in the lipid bilayer and a hydrophilic drug can be encapsulated in the aqueous core. The first liposome formulation we explored was a PROTAC-in-cyclodextrin liposome system (Figure 1). In the literature, drug-in-cyclodextrin liposomes have been shown to encapsulate hydrophobic drugs with low water solubility in the liposomal core [27]. Briefly, a PROTAC-in-cyclodextrin liposome was formulated, wherein the cyclodextrin was first incubated with the PROTAC to form a fully dissolved complex and this complex was then encapsulated into the aqueous core of the liposome.

The second system that we developed was a conventional liposome system, wherein the PROTAC was encapsulated into the lipid bilayer of the liposomes. Both liposome formulations were made using a microfluidic-based system, mixing organic and aqueous phases through custom-engineered microfluidic cartridges to obtain consistent high-quality liposomes.

For the PROTAC-in-cyclodextrin liposome, we purified the formulation by first centrifuging to remove any undissolved PROTAC, followed by dialysis to remove dissolved, but unincorporated, PROTAC and any residual organic solvent. For the conventional liposome, we utilized the Blaze microfluidic device to produce the liposome and TFF as purification techniques to remove any dissolved and unincorporated PROTAC and residual organic solvent.

Blaze and TFF are scalable processes that result in efficient, reproducible and high-quality liposomes with line of sight to clinical trials and beyond. Additionally, TFF is a technique to concentrate liposome formulations as well. To characterize and measure the loading and encapsulation efficiencies, an AKTA column separation was utilized. The AKTA-pure system can confirm liposome identity, remove impurities, and fractionate liposome-only formulations [28]. The liposome formulation was separated by particle size through the AKTA-pure system, where the first peak represented the liposomes and the second peak represented the unencapsulated free drug. The AUCs of these two peaks were utilized to determine the loading and encapsulation efficiencies of the liposome formulations (Equations (1) and (2)). As shown in Table 2, the encapsulation efficiencies of GNE-01 and GNE-02 in cyclodextrin liposomes were 28.1% and 16.5%, respectively.

The encapsulation efficiency of the conventional liposome formulation was optimized by altering the drug to lipid (D/L) ratio. We altered the D/L ratio stepwise from 0.017 to 0.2 and measured the EE using AKTA (Figure A1a). As the D/L ratio increased, more unencapsulated drug was present, decreasing the EE (Figure A1c). The highest EE was achieved with a D/L ratio of 0.017, indicating that the lipid bilayer was maximally encapsulated with the drug, with minimal free drug present. At a D/L of 0.017, the encapsulation efficiency was 68% and the loading efficiency was at 100%; hence, lower D/L ratios were not tested (Figure A1b,c). The D/L ratio of 0.017 was selected as the optimal ratio for the conventional liposome formulation for in vivo studies.

In these investigations, we discovered that it was simpler to control and optimize the EE for conventional liposomes where limited free PROTAC was present, compared to the PROTAC-in-cyclodextrin liposomes, where the EE remained relatively low. For the PROTAC-in-cyclodextrin liposome system, the PROTAC molecule was passively encapsulated into the aqueous core, resulting in an overall lower EE, compared to the conventional liposomes, wherein the PROTAC actively incorporated the lipids in the hydrophobic layer as liposomes were generated with the microfluidic device.

In addition to EE, particle size plays an important role in influencing the in vivo behavior within tumor microenvironments. These microenvironments have been shown to be heterogeneous, highly vascularized, and more permeable and leakier than normal tissues [29]. These features allow high molecular weight, non-targeted drug accumulation in tumors, which is known as the enhanced permeability and retention (EPR) effect [30]. There are several references in the literature that indicate that a particle size below 200 nm is amenable for passive targeting of tumor tissues via the EPR effect [30,31]. Particle size also has an impact on stability, encapsulation efficiency, drug release, bio-distribution, muco-adhesion and cellular uptake [32].

The Z-average particle size and polydispersity index (PDI) of the two types of liposomes are shown in Table 2. The PROTAC-in-cyclodextrin liposome particle size remained consistent between GNE-01 and GNE-02 at 149.0 nm and 128.1 nm, respectively. We also compared PROTAC-in-cyclodextrin liposome with the conventional liposome, and the particle size for both formulations remained similar, being 128.1 nm for cyclodextrin liposome and 121.0 nm for conventional liposome, which is consistent with previous results in the literature [33]. These sizes indicated that the additional incorporation of cyclodextrin did not affect the overall size of a liposome. All formulations prepared in this study showed optimum PDI below 0.2, which indicated good homogeneity and monomodal distribution of the liposomes.

Despite the similarity in particle size between these two liposomal drug formulations, notable differences were observed in their zeta potentials. The conventional liposome presented a lower absolute zeta potential value compared to the PROTAC-in-cyclodextrin liposome. This difference could, potentially, be attributed to the variation in pH between these two formulations. The PROTAC-in-cyclodextrin liposome was formulated in D5W solution at pH 7.4, while the conventional liposome was formulated in sodium acetate buffer at pH 4.0. In previous studies, it has been shown that lipids can significantly impact the zeta potential of liposomes, due to the extent of protonation of the primary amines on DSPC and DSPE-PEG2000 lipids, with more amines expected to be protonated at lower pH values, which could result in lower absolute zeta potential values [34].

With these optimized liposome formulations, a pharmacokinetic study was performed to understand the impact of liposomal formulation on the PROTAC molecules. The first PK study was performed with GNE-01 in a cyclodextrin liposome formulation. Sprague Dawley rats and SCID mice were divided into 2 cohorts and administered 1 mg/kg of either the GNE-01 PROTAC-in-cyclodextrin liposome or GNE-01 PROTAC in a solution formulation. We hypothesized that, if the PROTAC-in-cyclodextrin was encapsulated in the liposome core, then the PROTAC would be inaccessible to the general clearance mechanism of the soluble drug.

As a result, the plasma exposure of the liposome system should be enhanced, compared to the solution formulation. The plasma concentrations over time curves are shown in Figure 1a,b.

The solution formulations in both mice and rats showed the typical trend of an I.V. PK curve with rapid elimination of the drug upon administration. A significant increase was seen in the plasma concentrations when the PROTAC-in-cyclodextrin liposome drug delivery system was utilized with an 80-fold increase of AUC in mice, and a 380-fold increase in rats, compared to the standard I.V. solution formulation. To further model and understand the liposome interaction, a non-compartmental analysis was performed (Table 3). In addition to the enhancement in AUC, the liposome system significantly reduced the clearance of the PROTAC in both mouse (72 fold) and rat (342 fold) models, indicating a longer circulation for the PROTAC in the PROTAC-in-cyclodextrin liposome system. Overall, the PROTAC-in-cyclodextrin liposomal formulation showed enhancement in the PK parameters, compared to the solution formulation.

Based on the improved PK with the liposomal formulation, we further investigated the PROTAC-in-cyclodextrin liposome system in a PK/PD model in tumor-bearing mice. With the higher distribution of liposomes into tissues, we hypothesized that the PROTAC concentrations in the tumor would be enhanced with the PROTAC-in-cyclodextrin liposome system, compared to solution. We also hypothesized that the higher circulation times, along with higher tissue distribution, should improve accumulation into the tumor tissue.

The tumor PK data for GNE-01, seen in Figure 1c, generally agreed with our hypothesis. The accumulation, however, was not as pronounced as expected. Even though the plasma concentration of the PROTAC in the mouse PK study was 25× higher for the PROTAC-in-cyclodextrin liposome formulation over the conventional I.V. solution formulation (Table A1), this difference only translated to a 3× increase in the corresponding tumor concentration (Figure 1c). Furthermore, the target degradation, measured as % loss of BRM protein, was only 7% higher for the PROTAC-in-cyclodextrin liposomes, compared to the solution formulation (Figure 1d).

The PK/PD data suggests that the significantly higher PROTAC concentrations in the plasma did not translate to the same linear increase of PROTAC molecules in the tumor region. We hypothesized that this result could be due to poor and ill-defined vasculature in the tumor area and the potential for accumulation of the PROTAC-in-cyclodextrin liposomes in other organs, such as the liver and spleen. Additionally, even with the 3× increase of GNE-01 in the tumor observed with the PROTAC-in-cyclodextrin liposomes, there was no significant difference in the target degradation, compared to the solution formulation. This outcome indicated that the higher amount of PROTAC in the tumor did not necessarily translate to higher target engagement. One reason for this could be that the PROTAC molecules were still bound to the liposome carrier and, thus, not able to engage with the target.

To understand this disconnect we investigated two different attributes of the liposome formulation. First, we altered the location of the drug incorporated in the liposome. This was achieved by designing a conventional liposome formulation with enhanced encapsulation efficiency, wherein the hydrophobic PROTAC was incorporated in the lipid bilayer, instead of the liposome core. In this conventional liposome system, we hypothesized, the PROTAC molecules would release faster, as PROTAC molecules would need to only escape the lipid bilayer, as opposed to the hydrophilic core for PROTAC-in-cyclodextrin liposomes, wherein the PROTAC would need to fully cross the lipidic bilayer (Figure 1). Second, we developed an analytical method to track and correlate the lipids of the liposomes along with PROTAC in vivo. These studies were performed with GNE-02, which was similar to GNE-01 from the standpoint of the physicochemical properties of being highly lipophilic with poor water solubility. The motivation for switching to GNE-02 was partly due to the overall better drug-like properties, such as higher potency and better therapeutic index. The compound switch also gave us an opportunity to compare two similar PROTACs in our PROTAC-in-cyclodextrin liposome system to test the robustness of the formulation and the process system.

A mouse PK study was performed with three cohorts: GNE-02 PROTAC-in-cyclodextrin liposome, GNE-02 in conventional liposomes, and GNE-02 in conventional I.V. solution formulation (Figure 2). Similar to GNE-01, the PROTAC-in-cyclodextrin liposomes for GNE-02 showed an increase in AUC and a decrease in CL with a 23× enhancement and a 25× reduction, respectively, compared to the solution formulation (Table 4). These data strongly indicated that the strategy for enhancing PK properties with a PROTAC-in-cyclodextrin liposome system was consistent and might be applicable across multiple PROTAC compounds with similar physicochemical properties. The conventional liposome formulation for GNE-02 showed a modest increase in AUC of 5×, compared to the solution.

It is interesting to note that even though the conventional liposome enhancement was reduced, compared to the PROTAC-in-cyclodextrin liposomes, the PROTAC to lipid ratio in the plasma was significantly different for the conventional liposomes, compared to the PROTAC-in-cyclodextrin liposomes (Figure 2b). The PROTAC-in-cyclodextrin liposome had a constant PROTAC/lipid ratio over time, indicating that the liposomes were cleared at the same rate as the PROTAC molecule. We hypothesized that, as the liposome was cleared from the system, the PROTACs encapsulated within that liposome were cleared with it. This data strongly suggested that the PROTACs were still associated with the liposomes. In the meantime, in the conventional liposomes, the PROTAC/lipid ratio was not constant, and faster clearance of the PROTAC, compared to the lipid, was observed. This indicated that the PROTACs might be released from the liposome before the clearance of the liposome occurred. This observation suggested that the conventional liposome system might have faster release and, potentially, more free PROTACs available to engage the target protein.

Preliminary results from these simpler conventional liposomes indicated that the release kinetics could be improved, compared to the solution formulation. Additionally, progression of a simpler liposome formulation would have faster clinical development times, compared to a more complex cyclodextrin-based liposome system. Finally, it might be possible to further optimize these conventional liposomes by changing various parameters, such as the drug to lipid ratio, rate of mixing, and incorporation of other lipids etc., to improve target engagement that could result in superior efficacy over current formulations.

## 4. Conclusions

In this work we developed a foundation for the utilization of novel liposome systems to improve PROTAC systemic PK, exposure, and tumor accumulation relative to traditional solution-based formulations. The first PROTAC-in-cyclodextrin liposome formulations of GNE-01 had the unique attribute of encapsulating a high concentration of a hydrophobic API in the aqueous core. We also showed that this system is translatable to other PROTACs with similar physicochemical properties. The lack of improved BRM loss with the liposome system, over the conventional system, could be due to the slow release kinetics of the PROTAC from the PROTAC-in-cyclodextrin liposome. However, due to the inherent complexity with multiple variables in play, modifying the release kinetics of these liposome formulations proved to be difficult. In the meantime, we were able to optimize a simpler conventional liposome system with high PROTAC EE, wherein enhanced systemic circulation, and reduced clearance, were observed. Future studies with these conventional liposomes, to understand the release mechanism and rate of the PROTAC release from the liposome, would be important to understand their therapeutic efficacy.

## Data Availability

Raw data were generated at Genentech. Derived data supporting the findings of this study are available from the corresponding author V.S. on request.

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
