# Peer review of "Development of Liposome Systems for Enhancing the PK Properties of Bivalent PROTACs"

_pharmaceutics, 2023, doi:10.3390/pharmaceutics15082098_

Round 1
Reviewer 1 Report
This work reports the preparation of cyclodextrin-PROTAC liposome systems as a promising new technology in drug development. The document includes valuable material for publication and follows the journal's scope. The manuscript consists of physicochemical and biological characterization of the materials. However, some aspects need to be improved before being published. Here are the main observations:
1. Introduction. In the abstract, the work mentions that cyclodextrin-PROTAC liposome systems were developed; however, nothing is mentioned in the introduction regarding cyclodextrin, its important characteristics, and its advantages as an encapsulation system. In the same way, the objective of the work does not mention anything about this oligomer. Therefore, the authors must explain in a paragraph of the introduction why they are using cyclodextrin, highlighting its properties. Likewise, in the objective, they must explicitly state that they will make materials based on this oligomer. I suggest reviewing the introductory section of some references, where interesting aspects of cyclodextrin are highlighted, for example, https://doi.org/10.1007/s10965-020-02076-7
2. Page 3, line 108. The substance used to manufacture the formulations seems to be 2-Hydroxypropyl)-beta-cyclodextrin.
3. Justify using the 2-Hydroxypropyl)-beta-cyclodextrin derivative instead of pristine beta-cyclodextrin (introduction or results section).
4. Table 2. The authors mention that "each particle size measurement consisted of 15 acquisitions (page 4, line 152)"; Table 2 includes the values of PDI and hydrodynamic size. Please add the standard deviation of these measurements.
5. Page 8, lines 308-310. The authors mention that "Particle size also has an impact on stability, encapsulation efficiency, drug release, bio-distribution, mucoadhesion, and cellular uptake." In this sense, the zeta potential provides a more accurate estimate of the stability of the particles. Undoubtedly, the research acquired greater value if the measurement of the zeta potential of the systems studied was included. Likewise, studying the behavior of the hydrodynamic size with time would allow knowing the stability of the particles.
6. Page 8, lines 318. The authors mention, "All formulations prepared in this study showed optimum PDI below 0.2". Please explain why these values are considered optimal. Are they optimal for your study, or what do you mean? You can include a reference that supports your assertion.
7. Table 2. Samples GNE-01 and GNE-02 show different hydrodynamic size values; please explain what you attribute the difference in size to.
8Could you include SEM, AFM, or TEM images of liposome systems?
9 Figure 2 and Appendix A1 have a shallow resolution; the graphics and their legends are poorly perceived. Please add a sharp version.
The results section does not include an analysis of the graph in Fig A1(c).
The document contains some grammar and writing errors. Please thoroughly review the wording of the whole document.
Author Response
Point 1 : Introduction. In the abstract, the work mentions that cyclodextrin-PROTAC liposome systems were developed; however, nothing is mentioned in the introduction regarding cyclodextrin, its important characteristics, and its advantages as an encapsulation system. In the same way, the objective of the work does not mention anything about this oligomer. Therefore, the authors must explain in a paragraph of the introduction why they are using cyclodextrin, highlighting its properties. Likewise, in the objective, they must explicitly state that they will make materials based on this oligomer. I suggest reviewing the introductory section of some references, where interesting aspects of cyclodextrin are highlighted, for example, https://doi.org/10.1007/s10965-020-02076-7
Response 1:
We thank the author for bringing this to our attention. We have moved the section in the result and discussion where we introduced cyclodextrins now to the introduction. And we have also added more details elaborating on the system.
We have added the below sections in blue now to the intro to address this comment:
One strategy to enhance solubility of highly lipophilic molecules is utilizing cyclodextrins. Cyclodextrins have a unique cyclic structure that includes a hydrophobic core and a hydrophilic exterior. This design allows them to house hydrophobic molecules within the core, thus enhancing the water-solubility of the molecules that are typically water-insoluble [15, 16]. Numerous derivatives of cyclodextrins have been researched, each offering unique size, shape, and water solubility characteristics that result in various complexation interactions with drug molecules [17]. While cyclodextrins do improve the solubility, they generally do not alter the pharmacokinetics properties of the compound [18]. Therefore, aside from investigating cyclodextrins as a means to improve water solubility, it is crucial to develop alternative formulation techniques for improving PK properties of insoluble compounds.
Liposome drug carriers have been shown to improve both pharmacokinetic properties and efficacy of small molecule therapeutics [19]. Liposomes are very versatile and can encapsulate both lipophilic and hydrophilic therapeutics in the lipid bilayer and aqueous core, respectively. This encapsulation protects the drug during systemic circulation, thus increasing the systemic half-life, and improving therapeutic index [20]. In addition to traditional liposomes, cyclodextrin-in-liposome systems have been developed where the hydrophobic molecule localizes within the aqueous core of the liposome rather than in the lipid bilayer. Localizing the compound in different regions of the liposomes can result in different release profiles [21]. Nanoparticle carriers based on polymers and liposomes have been investigated as promising strategies to delivery PROTACs due to the lipophilic nature and required intracellular localization to generate a therapeutic response. [15, 22]. PROTACs are generally dosed I.V. due to their poor oral bioavailability so development of strategies to increase the systemic half-life can reduce the dosing frequency.
Here, we discuss the investigation of a conventional and a cyclodextrin based liposome system encapsulating the BRM target heterobifunctional PROTACs, GNE-01 and GNE-02 mainly to improve systemic clearance.
Point 2&3 : Page 3, line 108. The substance used to manufacture the formulations seems to be 2-Hydroxypropyl)-beta-cyclodextrin. Justify using the 2-Hydroxypropyl)-beta-cyclodextrin derivative instead of pristine beta-cyclodextrin (introduction or results section).
Response 2 & 3:
Thank you for bringing this to our attention, we have added motivation in the results section now to justify the use of HP-β-CD in these studies.
“Cyclodextrins have been developed with different derivatives that consist of varying physicochemical properties influencing their aqueous solubility and capacity. One of the common natural cyclodextrins is β-cyclodextrin. However, β-cyclodextrin is not suitable for parenteral formulations due to its limited solubility in water and associated toxicity [25]. In contrast, β-cyclodextrin derivatives, such as Sulfobutylether-β-Cyclodextrin (SBE-β-CD) and 2-Hydroxylpropyl-β-Cyclodextrin (HP-β-CD) possess higher aqueous solubility and have been successfully marketed in parenteral formulations, indicating a viable way forward with these derivatives.”
Point 4 : Table 2. The authors mention that "each particle size measurement consisted of 15 acquisitions (page 4, line 152)"; Table 2 includes the values of PDI and hydrodynamic size. Please add the standard deviation of these measurements.
Response 4:
The data was analyzed using Malvern zetasizer instead of Wyatt DynaPro Plate Reader III, the preparation method was updated in section 2.5 and the mean and standard deviation are also updated. We have also adjusted the mean and standard deviation in the table to not represent 15 acquisitions of the same sample but to represent the deviation of preparation of multiple liposome samples.
|
Compound |
Liposomes |
Z-Average particle diameter ± SD(nm) |
PDI ± SD |
Zeta Potential (mV) ± SD |
%EE |
|
GNE-01 |
Cyclodextrin |
149.0 ± 34.0 |
0.18 ± 0.01 |
Not Measured |
28.1 |
|
GNE-02 |
Cyclodextrin |
128.1 ± 23.1 |
0.19 ± 0.02 |
-30.36 ± 9.13 |
16.5 |
|
Conventional |
121.0 ± 9.7 |
0.17 ± 0.03 |
-7.45 ±1.44 |
68.0 |
One way ANOVA test indicates no significant difference between particle diameter of conventional & cyclodextrin liposomes
Point 5 : Page 8, lines 308-310. The authors mention that "Particle size also has an impact on stability, encapsulation efficiency, drug release, bio-distribution, mucoadhesion, and cellular uptake." In this sense, the zeta potential provides a more accurate estimate of the stability of the particles. Undoubtedly, the research acquired greater value if the measurement of the zeta potential of the systems studied was included. Likewise, studying the behavior of the hydrodynamic size with time would allow knowing the stability of the particles.
Response 5:
We have now included zeta potential measurements in this study and replaced the method of particle size, polydispersity index and zeta potential by zetasizer as mentioned in the previous comment. We have also added a discussion section
“Particle size distribution and polydispersity index (PDI) were characterized by dynamic light scattering (DLS) using Malvern Zetasizer Ultra (Malvern Instruments, UK). Liposomes were diluted 5 to 10 times depending on the starting concentration, into appropriate buffers and then 1 mL of sample was transferred to a cuvette for measurement. We also analyzed zeta potential based on the principles of laser Doppler velocimetry and electrophoretic mobility (EM) with the same instrument. All readings were taken at 25°C. Data analysis was performed by ZS Xplorer software.”
“Despite the similarity in particle size between these two liposomal drug formulations, notable differences were observed in their zeta potentials. The conventional liposome presented a lower absolute zeta potential value compared to the PROTAC-in-cyclodextrin liposome. This difference could potential be attributed to the variation in pH between these two formulation where the PROTAC-in-cyclodextrin liposome is formulated in D5W solution at pH 7.4, while the conventional liposome is formulated in sodium acetate buffer at pH 4.0. In previous studies, it has been shown that lipid can significantly impact the zeta potential of liposomes due to the extent of protonation of the primary amines on DSPC and DSPE-PEG2000 lipids, with more amines expected to be protonated at lower pH values, which could result in lower absolute zeta potential values [34].”
Point 6 : Page 8, lines 318. The authors mention, "All formulations prepared in this study showed optimum PDI below 0.2". Please explain why these values are considered optimal. Are they optimal for your study, or what do you mean? You can include a reference that supports your assertion.
Response 6:
PDI below 0.2 indicates that the particle size distribution is relatively narrow and uniform where most of the particles have a size close to the mean or median particle size. On the other hand, if the PDI is outside 0.2, either the particle size distribution is broad or with bi- or tri- modal distribution. Here are some references that also mention that PDI below 0.2 is considered a narrow particle size distribution.
Ref:
- https://doi.org/10.3390/nano8100847
- https://doi.org/10.1016/j.ijpharm.2018.11.060
Point 7 : Table 2. Samples GNE-01 and GNE-02 show different hydrodynamic size values; please explain what you attribute the difference in size to.
Response 7:
We thank the reviewer for this comment. What was originally reported was the mean from one sample measured 15 times with the DLS instrument. We have repeated this to attribute a standard deviation to multiple preparations of each sample. As can be seen from the table now, the difference in size is due to batch-to-batch variation. We believe there is no significant difference between the 2 liposome formulation sizes.
Point 8 : Could you include SEM, AFM, or TEM images of liposome systems?
Response 8:
This is a very good suggestion. During the time of this work our team did not have access to these imaging techniques. We are currently working to get access to these useful characterization tools. As we continue on our work with these liposome systems, we foresee using these imaging techniques to gain deeper understanding.
Point 9 : Figure 2 and Appendix A1 have a shallow resolution; the graphics and their legends are poorly perceived. Please add a sharp version.
Response 9:
Higher resolution figures have been uploaded.
Point 10 : The results section does not include an analysis of the graph in Fig A1(c).
Response 10:
Thank you for bringing this to our attention. We have added a sentence now to the results and discussion.
At D/L of 0.017the encapsulation efficiency was 68% and the loading efficiency was at 100%, hence lower D/L ratios were not tested (Fig. A1(b,c)).
Reviewer 2 Report
The manuscript describes a novel approach to solubilize and formulate PROTAC drugs for in-vivo applications using cyclodextrin mediated solubilization followed by liposomal delivery. The manuscript can be accepted for publication with the following suggested comments:-
1) Chemical structures of the PROTACs need to be included (unless it cannot be disclosed), indicating hydrophobicity of the molecule.
2) controlled release study of the liposomal formulation vs cyclodextrin solubilized PROTAC to really show the benefit of LNPs in sustained release.
Author Response
Point 1 : Chemical structures of the PROTACs need to be included (unless it cannot be disclosed), indicating hydrophobicity of the molecule.
Response 1 :
We agree with the reviewer that inclusion of the chemical structure would be very helpful in some of the discussions. The manuscript was initially written with the structure as part of the results and discussion. During the internal review process our legal team advised us to remove the structure and any language referring to the structure. This was one of the main stipulations for public disclosure of this work. The best we could do was include the measured Log P of the molecules in Table 1. These values provide a sense of the relative lipophilicity of the 2 molecules.
Point 2 : controlled release study of the liposomal formulation vs cyclodextrin solubilized PROTAC to really show the benefit of LNPs in sustained release.
Response 2 :
In vitro release systems for these nanoparticles have been shown to have limited correlation with the in vivo release due to the complexity seen in vivo compared to in vitro dissolution in a controlled environment. Parameters such as lipid diversity in systemic circulation and macrophages can influence release rates which are difficult to mimic in vitro (doi: 10.1016/j.jconrel.2015.09.052).
Due to the complexity, we determined that running in vivo studies with the two liposome formulations would generate a more translatable system when optimizing the formulations.
Reviewer 3 Report
The authors describe a novel method for encapsulation of Proteolysis-Targeting Chimeras using cyclodextrin, with further characterization of the produced system, and emphasis on its pharmacokinetic properties. Some comments need to be addressed before the manuscript can be accepted for publication:
1- Cyclodextrin is associated with safety-related concerns. Can the authors comment on the safety of the PROTAC-in cyclodextrin liposomes compared to conventional liposomes?
2- Since the described PROTACs are highly lipophilic, have the authors considered using more lipophilic lipids (such as ceramides) in addition to the proposed phospholipids to increase their encapsulation?
3- Please include the zeta potential values of the described PROTAC-in cyclodextrin liposomes and conventional liposomes, since surface charge of intravenously administered liposomes also dictate their PK behavior.
4- Table 2: Please report values as mean and S.D.
5- Statistical analysis is missing
Author Response
Point 1: Cyclodextrin is associated with safety-related concerns. Can the authors comment on the safety of the PROTAC-in cyclodextrin liposomes compared to conventional liposomes?
Response 1 :
The authors feel that safety would have to be evaluated in this system at some point. In general, some cyclodextrin molecules, when administered parenterally, have been associated with dose dependent toxicity, such as nephrotoxicity (https://doi.org/10.1111/j.2042-7158.2010.01030.x). Here with the PROTAC-in cyclodextrin liposomes system the cyclodextrin is entrapped within the liposome reducing the total amount of cyclodextrin in each dose compared to cyclodextrin formulations.
Additionally, when the cyclodextrin is entrapped in the core, we would expect minimal to no safety related finding as the cyclodextrin is not available to systemic circulation and tissues. As a result, we hypothesize that the total concentration of free cyclodextrin at a given time with this system would be minimal and probably safe. However, a safety study would be required to assess the safety of the cyclodextrin liposome system at varying doses.
Point 2 : Since the described PROTACs are highly lipophilic, have the authors considered using more lipophilic lipids (such as ceramides) in addition to the proposed phospholipids to increase their encapsulation?
Response 2 :
The authors thank the reviewer for the excellent suggestion. We did not consider other phospholipids for this work as this would have added another dimension/parameter to optimize and greatly increased the work and scope. The choice of DSPC as the lipid was mainly due to the universal acceptance of this lipid and regulatory approval on several liposomal products in the market.
Point 3 : Please include the zeta potential values of the described PROTAC-in cyclodextrin liposomes and conventional liposomes, since surface charge of intravenously administered liposomes also dictate their PK behavior.
Response 3 :
We have now included zeta potential measurements in this study (table 2) and replaced the method of particle size, polydispersity index and zeta potential by zetasizer.
Point 4 : Table 2: Please report values as mean and S.D.
Response 4 :
The data was analyzed using Malvern zetasizer instead of Wyatt DynaPro Plate Reader III, the preparation method was updated in section 2.5 and the mean and standard deviation were also updated. We have also adjusted the mean and standard deviation in the table to not represent 15 acquisitions of the same sample but to represent the deviation of preparation of multiple liposome samples.
|
Compound |
Liposomes |
Z-Average particle diameter ± SD(nm) |
PDI ± SD |
Zeta Potential (mV) ± SD |
%EE |
|
GNE-01 |
Cyclodextrin |
149.0 ± 34.0 |
0.18 ± 0.01 |
Not Measured |
28.1 |
|
GNE-02 |
Cyclodextrin |
128.1 ± 23.1 |
0.19 ± 0.02 |
-30.36 ± 9.13 |
16.5 |
|
Conventional |
121.0 ± 9.7 |
0.17 ± 0.03 |
-7.45 ±1.44 |
68.0 |
One way ANOVA test indicates no significant difference between particle diameter of conventional & cyclodextrin liposomes
Point 5 : Statistical analysis is missing
Response 5 :
We have now added the statistical analysis to the relevant graphs (Figure 1c and Table 3) and have added a section to the material and methods.
Statistical Analysis
Statistical analysis was performed using GraphPad Prism 8.0 Software (San Diego, USA). A t-test with Welch correction or one-way analysis of variance (ANOVA) was used to analyze significance between each group. Data is expressed as Mean ± standard deviation (SD) where * indicates p< 0.05 and ** indicates p< 0.01. A non-compartmental analysis was conducted utilizing the trapezoidal rule and relevant pharmacokinetics equations to calculate the pharmacokinetic parameters.
Round 2
Reviewer 1 Report
The most recent version of the manuscript has the necessary elements to be published in the Pharmaceutics journal.
Minor editing of English language required
Reviewer 3 Report
My comments have been addressed properly